# Potential Toxicity Risk Assessment and Priority Control Strategy for PAHs Metabolism and Transformation Behaviors in the Environment

**DOI:** 10.3390/ijerph191710972

**Published:** 2022-09-02

**Authors:** Lei Zhao, Mengying Zhou, Yuanyuan Zhao, Jiawen Yang, Qikun Pu, Hao Yang, Yang Wu, Cong Lyu, Yu Li

**Affiliations:** 1College of New Energy and Environment, Jilin University, Changchun 130012, China; 2MOE Key Laboratory of Resources and Environmental Systems Optimization, North China Electric Power University, Beijing 102206, China

**Keywords:** polycyclic aromatic hydrocarbons (PAHs), biological metabolism, environmental transformation, environmental risk, human health risk

## Abstract

In this study, 16 PAHs were selected as the priority control pollutants to summarize their environmental metabolism and transformation processes, including photolysis, plant degradation, bacterial degradation, fungal degradation, microalgae degradation, and human metabolic transformation. Meanwhile, a total of 473 PAHs by-products generated during their transformation and degradation in different environmental media were considered. Then, a comprehensive system was established for evaluating the PAHs by-products’ neurotoxicity, immunotoxicity, phytotoxicity, developmental toxicity, genotoxicity, carcinogenicity, and endocrine-disrupting effect through molecular docking, molecular dynamics simulation, 3D-QSAR model, TOPKAT method, and VEGA platform. Finally, the potential environmental risk (phytotoxicity) and human health risks (neurotoxicity, immunotoxicity, genotoxicity, carcinogenicity, developmental toxicity, and endocrine-disrupting toxicity) during PAHs metabolism and transformation were comprehensively evaluated. Among the 473 PAH’s metabolized and transformed products, all PAHs by-products excluding ACY, CHR, and DahA had higher neurotoxicity, 152 PAHs by-products had higher immunotoxicity, and 222 PAHs by-products had higher phytotoxicity than their precursors during biological metabolism and environmental transformation. Based on the TOPKAT model, 152 PAH by-products possessed potential developmental toxicity, and 138 PAH by-products had higher genotoxicity than their precursors. VEGA predicted that 247 kinds of PAH derivatives had carcinogenic activity, and only the natural transformation products of ACY did not have carcinogenicity. In addition to ACY, 15 PAHs produced 123 endocrine-disrupting substances during metabolism and transformation. Finally, the potential environmental and human health risks of PAHs metabolism and transformation products were evaluated using metabolic and transformation pathway probability and degree of toxic risk as indicators. Accordingly, the priority control strategy for PAHs was constructed based on the risk entropy method by screening the priority control pathways. This paper assesses the potential human health and environmental risks of PAHs in different environmental media with the help of models and toxicological modules for the toxicity prediction of PAHs by-products, and thus designs a risk priority control evaluation system for PAHs.

## 1. Introduction

Polycyclic aromatic hydrocarbons (PAHs), a class of persistent organic pollutants (POPs), consist of two or more mixed benzene rings [1,2]. Natural and artificial factors (i.e., volcanic eruptions, forest fires, biomass combustion, coke production, thermal processes, vehicle use, and waste combustion) significantly impact the generation of PAHs pollution in the environment [1,2]. Due to their low water solubility and persistent and semi-volatile properties [2,3], PAHs are easily accumulated and enriched in organism or different environmental media [4,5]. In addition, PAHs also have toxic, mutagenic, and carcinogenic impacts on organisms [6]. Specifically, PAHs can affect plants’ physiological processes (e.g., germination, growth, photosynthesis, and mineral absorption) [7]. PAHs can also result in sub-lethal damage to marine organisms [8] and endocrine disorders, immune toxicity and tumor lesions in human bodies [9]. The International Agency for Research on Cancer (IARC) has also identified 15 PAHs as potential carcinogenogens [10], among which Benzo [a] Pyrene (BaP) has been classified as a Class 1 carcinogen [11]. The United States Environmental Protection Agency (USEPA) has listed 16 kinds of PAHs as the priority control pollutants for remediation. PAHs have attracted significant attention due to their toxic harm to humans and the ecological environment [3].

PAHs have been widely detected in environmental mediums like water, the atmosphere, sediment, and the biological community [6,7]. Different forms of PAHs pollutants or their metabolites enter into soil, water, the atmosphere and other ecological environment media through evaporation, dissolution, precipitation, photo-oxidation, and biodegradation [12,13], affecting the survival and growth of plants and animals. The absorption and accumulation (biomagnification) of PAHs in the food chain results in potential carcinogenic, teratogenic, and mutagenic risks to humans [14]. Currently, there are many studies on the toxicity of PAHs. However, there are few reports on the toxicity risk of PAH metabolic intermediates, such as O-PAHs and OH-PAHs, decomposed or transformed in different natural environments [3]. The incomplete transformation of PAHs leads to the formation of some stable intermediates with even higher toxicity than the parent PAHs [14,15]. In addition, the presence of oxygen, nitrogen, or sulfur in PAH metabolites can also increase the toxicity of PAH derivatives. For example, 1-nitropyrene, a by-product of PYR, is more toxic than its parent [10].

The diversity of PAH derivatives [3] causes potential environmental and human health risks, therefore, the toxicity of PAHs and their derivatives should be considered comprehensively. Kouri et al. [16] tested BaP and its 13 derivatives for their ability to induce fibrosarcomas in inbred male mice. With the BaP dose of 0.9-μmol, it produced a 63% tumor incidence, while benzo[a]pyrene-7,8-oxide had very weak tumorigenic activity whereas the benzo[a]pyrene-4,5-, 9,10-, and 11,12-oxides had no tumorigenic activity. Masuda et al. [17] calculated the TEQ concentration of PAH and its derivatives in cooking exhaust gas and evaluated the exposure risk. Geier et al. [18] used high-throughput zebrafish assays to evaluate the morphological and neurobehavioral effects of 123 PAHs and their derivatives at concentrations ranging from 0.1 to 50μm. Liu et al. [19] evaluated the carcinogenic risk of PAHs and their derivatives during the dismantling process. However, present studies on PAH derivatives are limited to only single toxicity or specific production pathways; further comprehensive research is needed.

Traditional animal toxicity tests for all chemicals are not practical, therefore, computational toxicology based on computer technology has become an efficient tool for screening toxic compounds and assessing toxicity which can replace traditional toxicity testing methods in some cases [20]. Among the major in-silico approaches, quantitative structure-activity relationship (QSAR), Toxicity Prediction by Komputer Assisted Technology (TOPKAT), and VEGA were widely applied [21,22]. Sun et al. [23] established the QSAR model of acute oral toxicity of PAH rats by genetic algorithm and multiple linear regression methods. Zhong et al. [24] calculated Ames mutagenicity, carcinogenicity in rodents, oral median lethal dose in rats, and potential developmental toxicity with the help of the TOPKAT module, in order to comprehensively analyze the pharmacological properties and further screen suitable AKT1 drug candidates. Rioux et al. [25] employed the VEGA platform for predicting mutagenicity and carcinogenicity of the synthetized thiobarbituric acid derivatives. The toxicity risk of PAHs derivatives using the above toxicological methods has not been reported yet.

In this study, the transformation processes of PAHs and their derivatives in the environment and biological system were summarized. The potential risks of PAH derivatives in the various environment mediums were evaluated. A risk control evaluation system for PAHs was constructed based on the risk entropy method. The aims of this study were (1) to compare the by-product toxicity of different metabolism and transformation pathways of PAHs, (2) to evaluate and screen the metabolism or transformation pathway with the highest risk.

## 2. Materials and Methods

### 2.1. Environmental and Human Health Toxicity Characterization of PAHs

In this study, 16 types of priority control pollutant PAHs (Table 1) were taken as the research object. The toxicity variations of PAH by-products produced by the process of PAHs’ biological metabolism and environmental transformation were regarded as the environment and human health risk assessment indicators for PAHs and their derivatives. Among them, the PAHs’ potential environmental risk was characterized by phytotoxicity variations of PAHs and their derivatives. Human health risks were represented by neurotoxicity, immunotoxicity, genotoxicity, carcinogenicity, developmental toxicity, and endocrine disruption toxicity of PAHs and their derivatives. The neurotoxicity, immunotoxicity, and phytotoxicity activities of PAH derivatives were predicted using three-dimensional quantitative structure-activity (3D-QSAR) models. The developmental toxicity and genotoxicity of PAH derivatives were calculated by the Toxicity Prediction by Komputer Assisted Technology (TOPKAT) module in Discovery Studio 2020 (DS, Accelrys, Inc., San Diego, CA, USA) [26]. The carcinogenicity and endocrine-disrupting effects of PAHs and their derivatives were evaluated by VEGA platform [27].

### 2.2. Identification and Screening of Biological Metabolism and Environmental Transformation Products of PAHs

Pollution mediated by PAHs’ metabolism and transformation continues to threaten the environment, making this a priority issue [29,30]. This study reviewed the metabolism and transformation behavior of PAHs in different environmental media and organisms (organs) (Figure 1). The molecular structures of 16 PAH derivatives are given in Appendix A.

PAHs released into the environment change their state mainly depending on photoconversion or biodegradation [31,32]. Photodegradation is the main pathway for abiotic conversion of most PAHs in the environment [33]. After exposure to sunshine, skin-absorbed PAHs may lead to DNA damage under phototransformation (e.g., photodegradation and photooxidation) [32]. In addition, skin-absorbed PAHs can act as photosensitizers under the action of light. The intermediates of photo-oxidized PAHs tend to have the same or higher toxicity level than their parent compounds [32]; PAHs photosolution-related reaction system is prone to produce free radicals, which can induce biological function decline through oxidative stress [34].

Microorganisms, plants, and aquatic animals in the environment can degrade PAHs [35]. Microorganisms can convert PAHs into small compounds through respiratory reactions, fermentation, cometabolism, and dehalogenation [36]. In the process of aerobic degradation, oxygen contributes to the ring-opening reaction of PAHs and acts as the terminal electron acceptor; during anaerobic degradation, nitrogen, sulfur, metal ions (iron and manganese), carbon dioxide, chlorate, and trimethylamine oxide can replace oxygen as the terminal electron acceptor [4]. The degradation of PAHs by bacteria is mainly through the metabolic pathway mediated by oxygenase (including monooxygenase or dioxygenase) under aerobic conditions [37]. The aerobic bacteria first hydroxylate the aromatic rings of PAHs through dioxygenases to generate cis-dihydrodiols. Diol intermediates are then generated after dehydrogenases. These diol intermediates can be opened by ring-opening dioxygenases inside or outside the diol through ortho- or meta-cleavage to generate intermediates such as catechol, which are finally converted into tricarboxylic acid cycle intermediates [29]. Bacteria can also degrade PAHs through cytochrome P450-mediated pathways to produce transdihydrodiol, or metabolize PAHs under anaerobic conditions (such as nitrate reduction) [29]. Other pathways by which bacteria degrade BaP are also being discovered [38]. Fungi have been found to have the activity of synergistic metabolism for PAHs. Fungi can produce an extracellular lignin decomposition enzyme, which attacks PAHs with one or more peroxidase (POD enzyme). The intermediates are then metabolized by ring fission [15]. As an important primary producer, the metabolic pathway of PAHs in microalgae is still unclear, but dihydrodiol may be the first metabolite formed during PAHs degradation [39]. In addition, BaP can be oxidized by S. capricornutum to produce cis-4,5; 7,8; 9,10; 11,12-dihydrodiol-benzo[a]pyrene (BaP-73, BaP-74, BaP-75, BaP-76) [39,40].

Plants belong to autotrophs and are natural cleaning systems in the environment. Leaves, roots, and other parts of plants can absorb PAHs to complete plant metabolism for PAHs [41]. Plant metabolism is controlled by various enzymes (e.g., cytochrome P450 monooxygenase, POD enzyme, and polyphenol oxidase); thus, the pollutants in plants can be oxidized [41]. In addition, plants can remove PAHs from the soil through a series of bioinvertases secreted by their roots [35].

In animals and humans under the oxidative induction of cytochrome P450 monooxygenase group, PAHs can be biotransformed through an aryl hydrocarbon receptor (AHR) pathway. The corresponding epoxides are generated, some of which are highly active and can bind to DNA, leading to DNA damage and causing normal cells to turn into malignant cells, thus causing carcinogenic toxicity to the organism [29]. After exposure of PAHs to aquatic organisms for a long time, the detoxification of pollutants is related to the AHR gene deletion site. The decreased inducibility of the AHR signaling pathway may lead to the down-regulation of bioinvertase, especially CYP1A subtype enzyme [42]. When PAHs are metabolized in the human body, sites of epoxidation are initially transformed into phenols. If the polarity of PAHs is slight, it is directly combined with glucuronic acid or sulfuric acid, and finally discharged through feces or urine [43]. In addition, 3-hydroxy-benzo[a]pyrene (BaP-1) and other BaP metabolites have been found in human urine [43].

### 2.3. Neurotoxicity, Immunotoxicity and Phytotoxicity Prediction of PAHs and Their Derivatives Using a 3D-QSAR Model Method

#### 2.3.1. Molecular Structure Drawing of PAHs and Their Derivatives and Screening of Neurotoxic, Immunotoxic and Phytotoxic Receptor Proteins

The two-dimensional structures of PAHs and their derivatives were drawn by Chemdraw Ultra (Cambridge Soft, Boston, USA, 2010). The three-dimensional structures of PAHs were acquired through the ChemSpider (http://www.chemspider.com/) chemical structure database. The three-dimensional structures of PAH derivatives were drawn by Chemdraw 3D, and then geometrically optimized and energy-minimized using a molecular mechanics (MM2) force field. The minimum root-mean-square (RMS) gradient was set to 0.001 kcal/molÅ [44,45].

All protein crystal structures involved in this study were obtained from Protein Data Bank (PDB, https://www.rcsb.org/). Peroxidase (POD) plays an important role in plant resistance to environmental stress. Thus, POD (PDB ID: 1BGP, Appendix A) was selected as the phytotoxicity-related receptor of PAHs and their derivatives [46]. AHR mediates neurotoxic expression of PAHs [47,48], so AHR protein was selected as the neurotoxic receptor of PAHs (PBD ID: 5Y7Y, Appendix A). PAHs and their derivatives can cause cell apoptosis by activating the p53 signaling pathway related to cell growth and proliferation, thus expressing immunotoxicity. In this study, p53 protein (PDB ID: 2GEQ, Appendix A) was selected as the immunotoxic receptor of PAHs and their derivatives [47,49].

#### 2.3.2. Characterization of PAHs’ Neurotoxicity, Immunotoxicity and Phytotoxicity Effect: Molecular Docking and Molecular Dynamics Method

Before molecular docking, the receptor protein needed to be pretreated by deleting small ligand molecules and unnecessary water molecules in the protein. Polar hydrogen was added, missing side chains of protein were supplied, and redundant amino acids were deleted with the help of the “Protein Prepare” module via Discovery Studio 2020 (DS, Accelrys, Inc., San Diego, CA, USA). All the molecular docking parameters were optimized in DS. Firstly, docking receptors were defined in “Define Receptor”, and possible binding sites in receptor cavities were generated with the help of “From Receptor Activities” module, which was sorted by size. The pockets with the largest area and volume in the protein were considered as the most likely binding sites for ligands [50]. Therefore, the site selected as the binding site of the ligand with receptor and the docking radius was set to 10Å [51]. The docking conformation pattern was set as “Best”. Finally, the binding between receptor and ligand was realized in the Dock Ligands (LibDock) module. PAHs were docked to the neurotoxic receptor, immunotoxic receptor, and phytotoxic receptor, respectively, to obtain the PAHs-toxic receptor complex.

In this study, the binding free energy between PAHs and their toxic receptor was used as the activity parameter to characterize the toxicity of PAHs. With a Dell PowerEdge R7425 server, Groningen Machine for Chemical Simulation (GROMACS, Berendsen Laboratory, Göttingen University, Göttingen, Germany) 4.6.5 was applied to achieve the molecular dynamics simulations of PAHs and toxic receptor complex [52,53,54]. The molecular mechanics Poisson–Boltzmann surface area (MM/PBSA) method was used to calculate the binding free energy between PAHs and the toxic receptor complex [55] (Appendix A).

#### 2.3.3. Construction of 3D-QSAR Models for PAHs’ Neurotoxicity, Immunotoxicity and Phytotoxicity

In this study, the molecular dynamics simulation was carried out to calculate the binding free energy of PAHs and toxic receptors as the activity parameter to characterize the corresponding toxicity effect. Taking molecular structures of PAHs as independent variables and neurotoxicity, immunotoxicity, and phytotoxicity parameters as dependent variables, 3D-QSAR models for PAHs’ neurotoxicity, immunotoxicity, and phytotoxicity were constructed, respectively. The neurotoxicity, immunotoxicity, and phytotoxicity of PAH derivatives were predicted and evaluated by the 3D-QSAR model.

To ensure sufficient modelling data, in addition to the 16 priority pollutant PAHs, 7 additional kinds of PAHs [56] were also selected: 1-methylnaphthalene (1MNAP), 2-methylnaphthalene (2MNAP), 7H-benz[de]anthracen-7-one (benzanthrone, BANN), benzo[e]pyrene (BeP), 4H-cyclopenta (def)phenanthren-4-one (CPPHR), coronene (CRN), perylene (PRL). Therefore, 23 PAHs were used as training or test sets for model construction. The molecular quantity ratio of the training and test sets was 3:1. BaP was selected as the template, and the basic skeletons (benzene ring) of the training set and test set were super-combined through the “Align Database” module. The field parameters of the molecular stereoscopic field, hydrophobic field, electrostatic field, hydrogen bond acceptor, and donor field were calculated subsequently. Then, the leave-one-out (LOO) method and non-cross-validation regression (No Validation) method were respectively used for cross-validation analysis of the training set. The cross-validation coefficient Q^2^, the optimal principal component number N, the non-cross-validation coefficient R^2^, standard error of estimate (SEE) and test value F of the model were obtained [57]. In general, when the cross-validation coefficient Q^2^ > 0.5 and the non-cross validation coefficient R^2^ > 0.9 and close to 1, this indicated that the model has reliable prediction ability [58]. In addition, external verification of the constructed Comparative Molecular Similarity Index Analysis (CoMSIA) model was required with the help of test sets. The cross-validation coefficient R^2^_pred_ and standard error of predict (SEP) parameters were used to verify the external predictive ability of the model [59]. All the above 3D-QSAR modelling process was completed in SYBYL-X software.

### 2.4. Prediction of PAHs and Their Derivatives’ Genotoxicity and Developmental Toxicity Using TOPKAT Method

The genotoxicity and developmental toxicity of PAH derivatives were evaluated by using the TOPKAT module in Discovery Studio 2020, which was related to toxodynamic properties [26]. The TOPKAT module can calculate and verify compounds’ toxicity and environmental effects based on the two-dimensional structure of molecules [60]. In addition, it is necessary to determine whether the evaluated PAH derivatives are within the “With Optimal Prediction Space (OPS)” and “Within OPS Limits” of the model [61]. “Within OPS” is used to determine whether a small molecule is within the model’s optimal prediction space, and “Within OPS Limits” is used to measure whether a molecule is within the model’s optimal prediction space limits. If both are “True”, the result is credible; if the former is “False” and the latter is “True”, the results are relatively reliable; if both are “False”, the credibility of the results should be considered [61].

### 2.5. Prediction of PAHs and Their Derivatives’ Carcinogenicity and Endocrine Disruption Toxicity: Using VEGA Platform

Estrogen receptors were widely used as one of the molecular targets for screening endocrine disruptors [62]. Therefore, the affinity of PAHs and their derivatives with estrogen receptors were selected to characterize the endocrine-disrupting effect. The Carcinogenicity model (CAESAR) (version 2.1.9) and Estrogen Receptor Relative Binding Affinity model (IRFMN) (version 1.0.1) single-effect toxicity models were applied to predict the carcinogenicity and endocrine-disrupting effects of PAHs and its derivatives via VEGA platform v1.1.5 (http://www.vega-qsar.eu/), with the results exported in a .txt file. VEGA also provided an application domain (AD) index to evaluate whether the forecast satisfies the AD needs in an automatic cross-read approach [63], with several built-in parameters to evaluate the reliability of the forecast. VEGA divided the carcinogenicity and endocrine-disrupting effects of PAH derivatives into positive and negative activities. The reliability of the prediction levels include low reliability (compounds separated from AD), medium reliability (compounds may be separated from AD), and high reliability (compounds located in AD) [62].

### 2.6. Construction of a Priority Control Evaluation System for PAHs and Its Derivatives’ Toxicity Risk: Using Risk Entropy Method

The environmental impact and toxicological effects of PAHs and their derivatives, including neurotoxicity, oxidative stress, immunotoxicity, and genetic toxicity as well as reproductive and endocrine disruption effects have been confirmed [64]. In order to comprehensively and systematically measure the potential toxicity risk of PAHs and their derivatives, this paper intends to construct a set of priority-control evaluation systems based on the risk entropy method to evaluate the potential ecological risk of biodegradation and transformation behavior of PAHs in natural scenarios, and then prioritize the risks of different metabolic and transformation pathways. In the priority control evaluation system, the phytotoxicity of PAHs’ metabolism and transformation products were selected as environmental risk screening factors (A). Human health risk screening factors (B) included neurotoxicity, immunotoxicity, developmental toxicity, genotoxicity, carcinogenicity, and endocrine-disrupting effects. In contrast, bioconcentration factor (BCF) [65] and environmental persistence (half-life, persistence, and air half-life) were taken as environmental behavior screening factors (C). The scoring rules of the optimal control evaluation system for a total risk of PAHs’ metabolism and transformation pathway exposure were as follows: for the neurotoxic, immunotoxic, phytotoxic, and environmental persistence (half-life, persistence, and air half-life) factors, when the predicted toxicity or environmental persistence of PAH derivatives was lower than that of their parent PAHs, the risk was negligible, and 0 was scored; when the parameter increases by 0–50%, the score was recorded as 1; when the parameter increases by more than 50%, the score was recorded as 2. For developmental toxicity and genotoxicity, the predictive probability value of low probability (0.0–0.30), medium probability (0.30–0.70) and high probability (>0.70) corresponded with 0, 1, and 2 points, respectively. When the predicted carcinogenicity and endocrine-disrupting effects of PAH derivatives were inactive, the score was 0, while the predicted carcinogenicity and endocrine-disrupting effects were active, the score was recorded as 1 point. For bioenrichment, when logBCF < 2.0 [66], the score was 0; when 2.0 ≤ logBCF < 3.0, the score was 1; and when logBCF ≥ 3.0, indicating the strong bioenrichment ability of the molecule [67], the score was 2. According to Formula (1), the total exposed risk assessment value $RA_{i}$ of PAHs’ metabolism and transformation pathway was calculated.
(1)$$RA_{i}=A_{i}+B_{i}+C_{i}=a_{i}+\frac{\sum_{j=1}^{n}b_{ij}}{6}+\frac{\sum_{j=1}^{n}c_{ij}}{4}$$

In the formula, $RA_{i}$ indicated the total exposed risk evaluation value of the *i*_th_ metabolic or transformation pathway of PAHs. $A_{i}$ referred to the environmental risk screening factor value for the *i*_th_ metabolic or transformation pathway of one of the PAHs. $B_{i}$ referred to the human health risk screening factor value for the *i*_th_ metabolic or transformation pathway of one of the PAHs. $C_{i}$ referred to the environmental behavior screening factor value for the *i*_th_ metabolic or transformation pathway of one of the PAHs. $A_{I}$ referred to environmental risks (i.e., phytotoxicity) evaluation value of PAH derivatives in the *i*_th_ pathway of one of the PAHs. $b_{\mathrm{ij}}$ represented the risk evaluation value of the *j*_th_ factor in human health risk screening factors for the *i*_th_ pathway of one of the PAHs. $c_{\mathrm{ij}}$ represented the risk evaluation value of the *j*_th_ factor in environmental behavior screening factors for the *i*_th_ pathway of one of the PAHs.

## 3. Results and Discussion

### 3.1. Prediction for Neurotoxicity, Immunotoxicity and Phytotoxicity of PAH Derivatives Based on 3D-QSAR Models

#### 3.1.1. Construction and Evaluation of 3D-QSAR Models for PAHs’ Neurotoxicity, Immunotoxicity and Phytotoxicity

In this paper, 3D-QSAR models were constructed to predict the neurotoxicity, immunotoxicity, and phytotoxicity of PAHs and their derivatives. As the parameters of the 3D-QSAR model are shown in Table 2, the 3D QSAR model for PAHs’ neurotoxicity performed well with prediction ability, with Q^2^ of 0.749 (>0.5) and R^2^ of 1.000 (>0.9). Additionally, the low SEE, high F and high R^2^_pred_ (>0.6) indicated that the constructed model had good fitting ability and robustness [68,69], suitable for applying in the evaluation of PAH derivatives’ neurotoxicity.

The test set was used to prove the constructed models possessed ideal internal and external validation parameters, indicating that the 3D-QSAR model expressed good evaluation stability and prediction ability. Experimental values, predicted values, and relative errors of the molecules in the 3D-QSAR model are shown in Table 3.

#### 3.1.2. Prediction of PAH Derivatives’ Neurotoxicity, Immunotoxicity, and Phytotoxicity

PAH derivatives’ neurotoxicity, immunotoxicity, and phytotoxicity were predicted based on the constructed 3D-QSAR models. The results are shown in Appendix A. Among them, the benzene rings of NAP-19, NAP-20, NAP-21 and NAP-22 had been cracked and no longer had the basic ring skeleton structure as PAH derivatives. Thus, the phytotoxicity, neurotoxicity, and immunotoxicity of these molecules could not be predicted using the constructed 3D-QSAR model.

Changes in toxicity parameters of PAH derivatives compared with their precursors were used to characterize the corresponding toxicity intensity. In Appendix A, PAH derivatives, except for ACY, CRH, and DahA, have higher neurotoxicity corresponding to their precursors. Selecting BaP as an example, whose derivatives’ neurotoxicity varied from 43.05% to 64.92%, BaP-51 was the most neurotoxic one, with increased neurotoxicity of 64.92% compared with parent BaP. BaP-51 (phenacemide) can be used as an anticonvulsant and cause acute psychosis [70].

Among the 473 PAH derivatives, 152 have higher immunotoxicity than their precursors, with an increase range of 0.52–674.54%. Only the immunotoxicity of all DahA derivatives decreased. A total of 222 PAH derivatives showed higher phytotoxicity than the precursors, whose toxicity increased by 0.03–527.74%. Only the phytotoxicity of BbF derivatives decreased. After the metabolic and transformation process, BghiP-6 produced by biological (rat) metabolisms exhibited the highest neurotoxicity, immunotoxicity, and phytotoxicity with an increased rate of 593.91%, 674.54%, and 527.74%, respectively.

### 3.2. Prediction of PAH Derivatives’ Developmental Toxicity and Genotoxicity Based on TOPKAT Method

The Developmental Toxicity Potential (DTP) (version 3.1) model in the TOPKAT module was applied to predict PAH derivatives’ developmental toxicity via DS. In contrast, the Ames Mutagenicity (version 3.1) model was accepted to predict the genotoxicity of PAH derivatives (Appendix A).

In Appendix A, excluding ACE, ACY, BghiP, FLR, and IcdP, 155 derivatives derived from the rest of PAHs possessed potential developmental toxicity. These derivatives contain 32 BaP derivatives, 17 BaAN derivatives, 26 FRT derivatives, 29 PHE derivatives and 16 PYR derivatives. The genotoxicity of 138 PAH derivatives increased compared with that of PAHs; thus, the metabolism and transformation of PAHs might lead to genotoxicity risk. ANT is not genotoxic [71]; however, its derivatives have increased genotoxicity.

### 3.3. Prediction of PAH Derivatives’ Carcinogenicity and Endocrine-Disrupting Effect Based on VEGA Platform

The carcinogenicity of PAH derivatives was predicted by the Carcinogenicity model (CAESAR) (version 2.1.9) in the VEGA platform. The endocrine disruption activity of PAH derivatives was predicted through the Estrogen Receptor Relative Binding Affinity Model (IRFMN) (version 1.0.1) model in the VEGA platform (Appendix A). The prediction ability of these two models for PAH derivatives were classed as “low reliability”, “moderate reliability”, “good reliability”, and “experimental value”, respectively.

According to the prediction results in Appendix A, among 473 PAHs metabolism and transformation derivatives, 247 PAH derivatives were active in carcinogenicity. Only the transformation products of ACY in nature do not have carcinogenicity. Taking BaP, a class I carcinogen, as an example, its 54 BaP derivatives retained the carcinogenic activity. This finding further highlights BaP carcinogenicity concerns that arise from frequent incidental exposures [72]. In addition to ACY, the 15 PAHs produced 123 endocrine-disrupting substances during the metabolism and transformation process, among which 29 BaP derivatives were included.

### 3.4. Potential Human and Environmental Health Risk Evaluation of PAH Derivatives

For 16 PAHs, the corresponding toxicity risk of PAH derivatives was evaluated based on the toxicity change of PAH derivatives compared with the parent PAHs. The potential environmental risk (phytotoxicity) and human health risk (neurotoxicity, immunotoxicity, developmental toxicity, genotoxicity, carcinogenicity, and endocrine-disrupting effect) of PAH derivatives produced in different metabolism and transformation behavior were evaluated and compared (Appendix A). In this study, the proportion of derivatives with increased toxicity was used to represent the risk probability of corresponding toxicity. In contrast, the molecular parameters’ average variation degree of those derivatives with a particular toxicity risk was used to represent the risk degree of this toxicity. The risk probability and degree of metabolism and transformation pathways of 16 PAHs are visualized in Figure 2. It can be seen that the priority-controlled PAHs (except ACY, CHR, and DahA) expressed potential neurotoxic risks. The PAHs (except DahA) after metabolism and transformation behavior had immunotoxicity risks. There was potential developmental toxicity risk in the metabolic and transformation behaviors of some priority-controlled PAHs (except ACE, ACY, BghiP, FLR, and IcdP). The biodegradation and environmental transformation pathways of ACE, ACY, ANT, FLR, NAP, PHE, and PYR lead to genotoxicity risks. The selected PAHs (except ACY) caused carcinogenic and endocrine disturbance risks. Potential environmental (phytotoxicity) threats in PAHs (excluding BbF) were found after metabolism and transformation pathways.

As shown in Figure 2, the metabolism and transformation behaviors of 16 kinds of PAHs had potential environmental and human health risks. It was obvious to identify the difference in the risk probability and degree in different PAHs, different pathways, or different toxicity risks. Taking BaP as an example, the metabolism and transformation products of BaP also had a relatively high probability of potential environmental (phytotoxicity) and human health (neurotoxicity, immunotoxicity, developmental toxicity, carcinogenicity, and endocrine disruption effect) risks. BaP derivatives were the same as BaP in terms of carcinogenic effects [64]. The high lipophilicity of PAHs increases their bioavailability in organisms, further increasing their carcinogenic risk [73]. The comparative products of BaP for human metabolism were biological (fish) metabolisms, which had 100% immunotoxicity risk, 100% developmental toxicity risk, 100% carcinogenic risk and 100% endocrine-interference risk. Compared with the parent BaP, the immunotoxicity and developmental toxicity of BaP derivatives generated in a human metabolism (or biological (fish) metabolism) pathway increased by 19.32% on average and 0.02 on average, respectively, and all of the derivatives in these two pathways were carcinogenic and endocrine-disrupting. Photolysis made BaP potentially harmful to the human body and environment (neurotoxicity, immunotoxicity, phytotoxicity, developmental toxicity, carcinogenicity, and endocrine-disrupting effect). The BaP derivatives produced by the photocatalytic degradation pathway presented neurotoxicity, immunotoxicity, phytotoxicity, developmental toxicity, carcinogenesis, and endocrine disturbance risks, with the neurotoxicity risk in this pathway (probability 45.45%, degree 27.42%) higher than in other pathways, with a higher risk degree of immunotoxicity (33.27%) and risk probability of potential phytotoxicity (72.73%). BaP derivatives derived from plant-defense degradation had a 20% probability risk of neurotoxicity (increased by 18.48%), 40% probability risk of immunotoxicity (increased by 22.08%), 80% probability risk of developmental toxicity (increased by 0.02%-degree increase), and 100% probability carcinogenic and endocrine disruption risk. There were potential (probability 33.33%) neurotoxicity risk (increased by 18.21%), potential (probability 33.33%) immunotoxicity risk (increased by 19.32%), potential (probability 33.33%) phytotoxicity risk (degree 2.61%), 100% probability developmental toxicity, carcinogenicity, and endocrine disruption risk. There was a 22.22% probability of potential neurotoxicity (degree 14.48%) risk, 33.33% probability of potential phytotoxicity (degree 19.69%) risk, and a 55.56% probability of potential carcinogenicity in the aerobic degradation pathway. Microbial (bacteria) degradation presented the risk of neurotoxicity, immunotoxicity, phytotoxicity, developmental toxicity, carcinogenicity, and endocrine disruption risk. Microbial (microalga) degradation of BaP had a 50% probability of potential immunotoxicity risk (10.28%-degree increase), 100% probability of developmental toxicity risk (0.02-degree increase), 100% probability of carcinogenicity risk, and a 50% probability of endocrine disruption effect. From the perspective of reaction pathways, only the immunotoxicity and developmental toxicity of BaP derivatives in the aerobic degradation pathway were lower than the parent BaP, and there was no endocrine disruption effect in risk evaluation for the aerobic degradation of BaP.

The toxicity risk of PAHs in the environment was characterized using the degree and probability of toxicity of PAH derivatives. Supposing there is no potential toxicity risk of a PAH derivative, the probability of this derivative is 0%, and the toxicity risk of PAHs could be ignored (i.e., the risk degree was regarded as 0% in default). Finally, the results of risk assessment for environmental (phytotoxicity) and human health (neurotoxicity, immunotoxicity, developmental toxicity, genotoxicity, carcinogenicity, and endocrine-disrupting effect) of PAHs were visualized in Figure 3a. It shows that BghiP has a significant neurotoxic impact, while Figure 3c shows that BkF and BghiP presents significant phytotoxicity risks. Figure 3f and Figure 4g show that the carcinogenicity and endocrine-disrupting effect of PAH derivatives remained widespread as the parent PAHs. According to Figure 3h, BghiP and BkF derivatives presented a high degree of toxicity risk with low-risk probability. Therefore, the overall risk ranking of 16 PAHs needed to be further evaluated and judged.

As POPs pollutants, PAHs lead to risks during the metabolism and transformation behaviors in the environment and organisms. It is necessary to strengthen the monitoring of PAHs and their derivatives in the environment to better identify the risk level of PAHs and their derivatives. PAHs were mainly distributed in tissues after being absorbed by organisms. Due to differences in tissue distribution, the toxic expression of PAHs depends to some extent on the ability of these organs to metabolize and detoxify PAHs [74]. Therefore, in the process of actual pollution remediation and risk control, the main components and transmission modes of PAHs pollution were analyzed specifically to adopt appropriate coping strategies.

In addition, the interaction between PAHs and their derivatives were not considered in the toxicity evaluation for degradation, metabolism, and transformation products of PAHs, which may have caused some deviation in the predicted results. Human health risks caused by PAHs were also affected by multiple factors such as the sex of the organism, exposure time, and dose [73].

### 3.5. Construction of a Priority Control Evaluation System of Total Exposed Risk of PAHs’ Metabolism and Transformation Pathway Based on the Risk Entropy Method

Due to the abundant types and quantities of PAHs metabolism and transformation products in the environment, the damage degree of PAHs to the ecological environment and human health varied greatly [3,75]. The toxicity equivalent method was usually used to evaluate PAHs’ risks in most cases. However, a comprehensive approach for assessing the potential risk level of PAHs exposed to the environment has barely been studied before [76]. Therefore, this paper constructed a priority control evaluation system based on the risk entropy method to evaluate the potential ecological risks of PAHs’ biodegradation and transformation behavior in natural scenarios, by which the metabolic and transformation pathway risks were ranked. Based on the above ranking results, it will be a scientific strategy to effectively solve environmental problems and prioritize the pollutants that significantly harm human health and the environment.

According to PAHs’ total exposed risk evaluation value RA (Figure 4), the total exposed risk of 16 PAHs in the natural environment was ranked as BaP, PHE, FRT, NAP, PYR, IcdP, FLR, ANT, ACE, BbF, BaAN, BghiP, BkF, CHR, ACY, and DahA.

Based on the total exposed risk evaluation value $RA_{i}$ (in Appendix A) of metabolic and transformational behavior, the pathways of 16 PAHs in the natural environment were ranked. After assessing the microbial (bacteria) degradation pathway, the total risk value of PAHs followed the reduced sequence of BaP, FRT, PHE, ACE, PYR, FLR, IcdP, NAP, ANT, BkF, ACY, BbF, and CHR. The total exposed risk evaluation value $RA_{i}$ of microbial (bacteria) degradation pathways of BaP, FRT, PHE, ACE, PYR, FLR, IcdP, NAP, ANT, BkF, and ACY were higher than the user-defined risk threshold (the pathway with the total exposed risk evaluation value in the top 30% [77] was divided into high-risk level), accounting for the necessity of those 11 PAHs’ priority control. In the risk assessment for human metabolism, the PAH sequence was NAP, BaAN, BbF, PHE, PYR, FLR, BkF, CHR, BaP, IcdP, and FRT according to the total exposed risk assessment value from high to low. Only NAP had a high total exposed risk to human health. For the biological (rat) metabolism, the risk of metabolic BghiP derivatives needed to be identified first. In the photolysis risk assessment, this pathway’s total exposed risk of BaP should be prioritized for prevention and control. The risk of the photocatalytic degradation pathway of BaP and PYR was classed as high. For biological (microalgae) metabolic behavior, PHE could be considered a priority control molecule. Biological (fungi) degradation pathways of FRT and ANT needed to be prioritized for prevention and control. The biological (mouse) metabolic pathways of BbF and IcdP were listed as risk priority control pathways. In aerobic degradation, the toxicity risk of BaP and NAP derivatives deserved attention. The risk of the biological (fish) metabolic pathway of FRT needed to be priority-controlled. The total exposed risk of other degrading, metabolic, and transformation pathways was moderate. It could also be seen that the toxicity risk of PAHs was not highly correlated with its bioaccumulation and absorption pathways, which had been proved by previous studies [74].

In the practice of risk prevention and control, the risk priority control evaluation system significantly reduces the uncertainty of comprehensive risk evaluation. It is essential to monitor and control high-risk pollutants during their metabolism or transformation behaviors [78]. These priority control pathway results may be used to carry out risk management for pollutants and improve management capacity, to as well as provide effective and targeted direction for decision makers [79].

## 4. Conclusions

Taking 16 PAHs priority control pollutants as the research objects, the phytotoxicity, neurotoxicity, immunotoxicity, developmental toxicity, genotoxicity, carcinogenicity, and endocrine-disrupting effect of PAH derivatives produced during metabolism and transformation were predicted. The potential environmental and human health risk caused by PAHs’ metabolism and transformation were evaluated, providing a theoretical basis and technical support for the risk assessment of PAHs and their derivatives in the real environment. The specific conclusions of this paper were as follows:(1)The neurotoxicity, immunotoxicity, and phytotoxicity 3D-QSAR models were constructed separately. The evaluation parameters indicated the good evaluation stability and prediction ability of the three models.(2)Biological metabolism and environmental transformation behavior of 16 PAHs in the environment were summarized, and 473 PAH derivatives generated by different pathways were summarized.(3)The 3D-QSAR model, TOKPAT, and VEGA platform were used to predict the phytotoxicity, neurotoxicity, immunotoxicity, developmental toxicity, genotoxicity, carcinogenicity, and endocrine-disrupting effect of PAHs and their derivatives. The environmental toxicity (phytotoxicity) and human health toxicity (neurotoxicity, immunotoxicity, developmental toxicity, genotoxicity, carcinogenicity, and endocrine-disrupting effect) of PAH derivatives generated in different metabolic and transformation pathways were comprehensively and systematically predicted.(4)The potential risks (environmental and human toxicity risk) of the metabolic and transformation behaviors of the 16 PAHs were evaluated. The derivatives of BghiP produced in rat metabolisms presented a high degree of neurotoxicity risk (risk degree 179.76%), immunotoxicity risk (degree 369.49%), phytotoxicity risk (degree 240.50%), and potential carcinogenic and endocrine-disrupting risk. According to the risk assessment results, partial PAH derivatives should also be included in the critical monitoring objects to realize the risk control of the whole life cycle of PAHs in the environment.(5)The risk priority control evaluation system for PAHs based on the risk entropy method was constructed, which was of great significance for screening the priority control pathway of PAHs in the environment.

## Figures and Tables

**Figure 1 ijerph-19-10972-f001:**
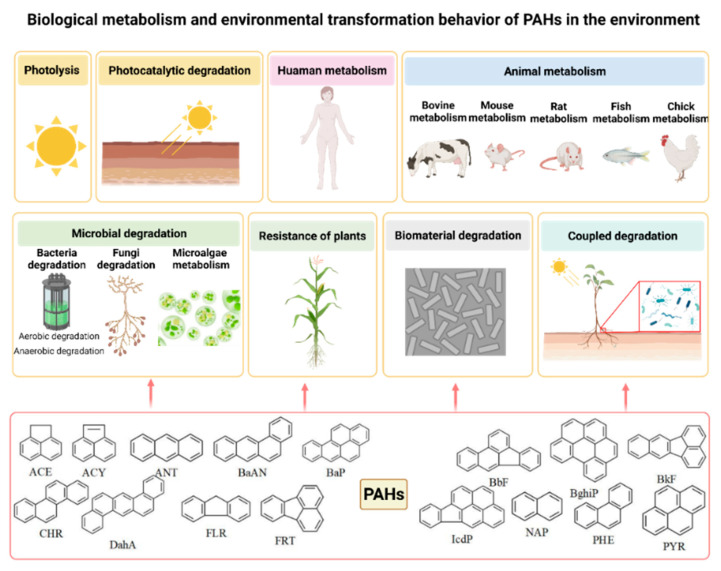
Schematic diagram of biological metabolism and environmental transformation of PAHs in the environment.

**Figure 2 ijerph-19-10972-f002:**
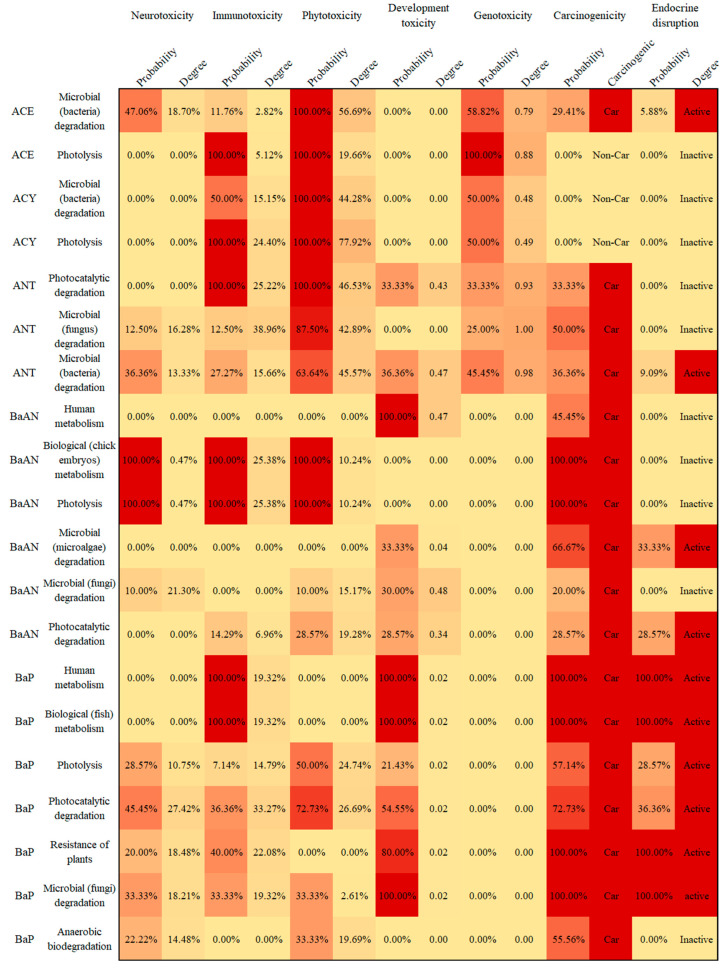
Risk diagram of biological metabolism and environmental transformation of 16 PAHs. (The intensity of the color indicates the enhanced intensity of toxicity risk probability or degree (The enhanced intensity of reduced toxicity reduction was regarded as 0)).

**Figure 3 ijerph-19-10972-f003:**
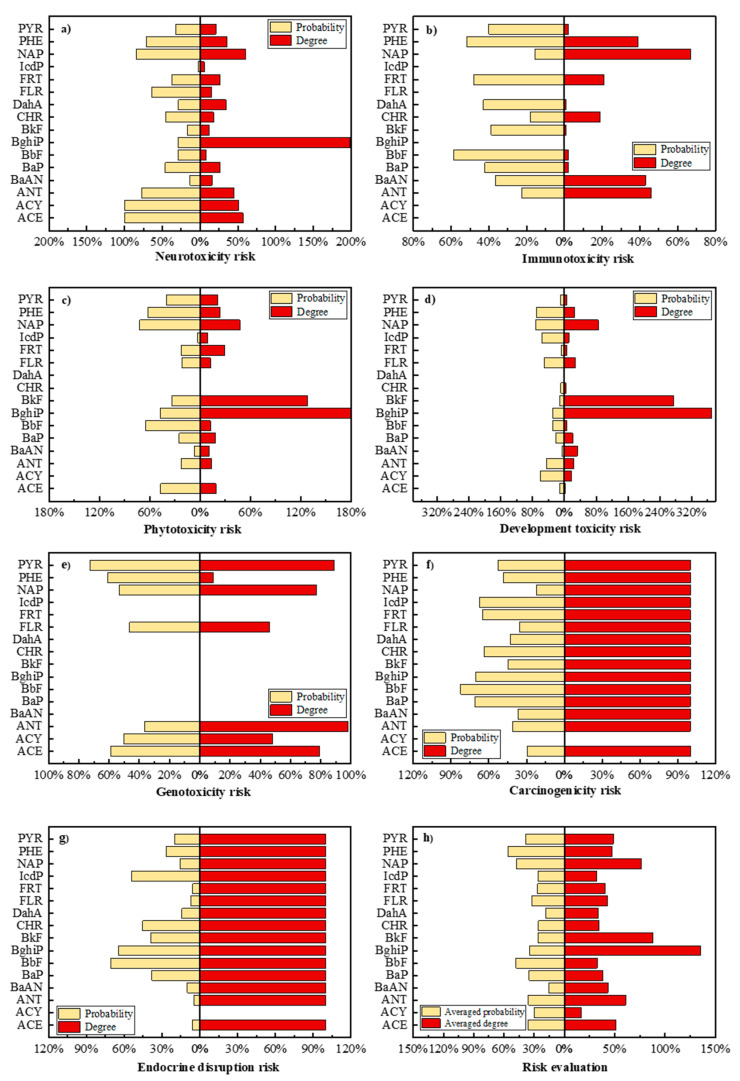
Comparison of toxicity risk (probability and degree) of 16 PAHs. (**a**). Comparison of neurotoxicity toxicity risk (probability and degree) of 16 PAHs; (**b**). Comparison of immunotoxicity risk (probability and degree) of 16 PAHs; (**c**). Comparison of phytotoxicity risk (probability and degree) of 16 PAHs; (**d**). Comparison of development toxicity risk (probability and degree) of 16 PAHs; (**e**). Comparison of genotoxicity risk (probability and degree) of 16 PAHs; (**f**). Comparison of carcinogenicity risk (probability and degree) of 16 PAHs; (**g**). Comparison of endocrine disruption risk (probability and degree) of 16 PAHs; (**h**). Risk (averaged probability and degree) evaluation of 16 PAHs).

**Figure 4 ijerph-19-10972-f004:**
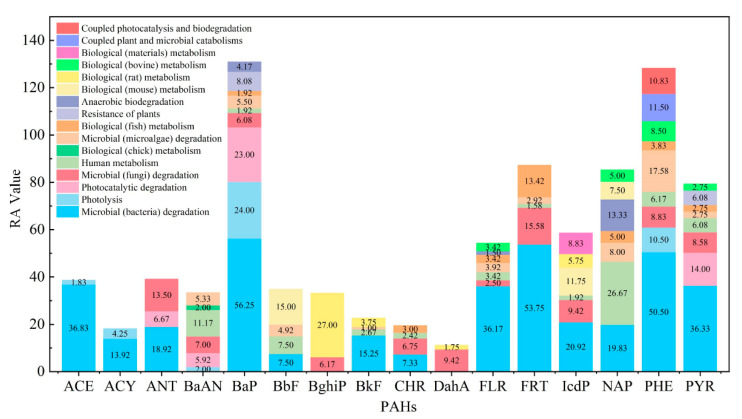
Total exposed risk evaluation value of degradation and transformation of PAHs.

**Table 1 ijerph-19-10972-t001:** Basic information of 16 priority pollutants of PAHs.

PAHs	Acronym	2D Structure	IACR Classifications Group [28]
Acenaphthene	ACE	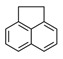	3
Acenaphthylene	ACY	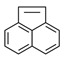	*
Anthracene	ANT	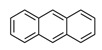	3
Benz[a]anthracene	BaAN	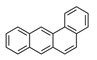	2B
Benzo[a]pyrene	BaP	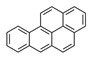	1 ^a^
Benzo[b]fluoranthene	BbF	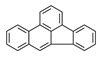	2B
Benzo[ghi]perylene	BghiP	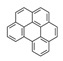	3
Benzo[k]fluoranthene	BkF	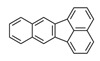	2B
Chrysene	CHR	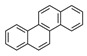	2B
Dibenz[a,h]anthracene	DahA	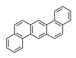	2A ^b^
Fluorene	FLR	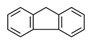	3
Fluoranthene	FRT	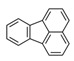	3
Indeno[1,2,3-cd]pyrene	IcdP	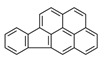	2B ^c^
Naphthalene	NAP	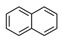	2B
Phenanthrene	PHE	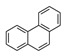	3 ^d^
Pyrene	PYR	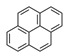	3

Note: * not classified by the IARC Monographs; ^a^ carcinogenic to humans; ^b^ probably carcinogenic to humans; ^c^ possibly carcinogenic to humans; ^d^ not classifiable as to its carcinogenicity to humans.

**Table 2 ijerph-19-10972-t002:** Evaluation parameters of 3D-QSAR models for PAHs’ neurotoxicity, immunotoxicity and phytotoxicity.

3D-QSAR Models	Q^2 a^	N ^b^	R^2 c^	SEE ^d^	F ^e^	R^2^_pred_ ^f^
PAHs’ neurotoxicity model	0.749	10	1.000	0.387	2370.819	0.789
PAHs’ immunotoxicity model	0.753	10	1.000	0.685	1991.539	0.639
PAHs’ phytotoxicity model	0.935	10	1.000	0.367	5032.620	0.650

Note: ^a^ optimum number of components obtained from cross-validated; ^b^ cross-validated determination coefficient; ^c^ non-cross-validated determination coefficient; ^d^ standard error of the estimate; ^e^ Fischer value; ^f^ external validation determination coefficient.

**Table 3 ijerph-19-10972-t003:** Experimental values, predicted values and relative errors of the molecules in training set and test set of models for PAHs’ neurotoxicity, immunotoxicity, and phytotoxicity.

PAHs’ Neurotoxicity Model				PAHs’ Immunotoxicity Model				PAHs’ Phytotoxicity Model			
PAHs	Exp. (Binding Free Energy, kJ/mol)	Pred. (Binding Free Energy, kJ/mol)	Relative Error (%)	PAHs	Exp. (Binding Free Energy, kJ/mol)	Pred. (Binding Free Energy, kJ/mol)	Relative Error (%)	PAHs	Exp. (Binding Free Energy, kJ/mol)	Pred. (Binding Free Energy, kJ/mol)	Relative Error (%)
1MNAP ^b^	−43.422	−47.198	−8.70	1MNAP ^b^	−61.808	−41.487	32.88	1MNAP ^a^	−37.273	−37.162	0.30
2MNAP ^b^	−46.392	−47.670	−2.75	2MNAP ^a^	−47.673	−47.812	−0.29	2MNAP ^a^	−42.014	−42.319	−0.73
ACE ^a^	−32.347	−32.025	1.00	ACE ^a^	−68.893	−68.577	0.46	ACE ^a^	−54.961	−54.710	0.46
ANY ^a^	−43.915	−44.111	−0.45	ACY ^a^	−53.244	−53.560	−0.59	ACY ^a^	−53.263	−53.552	−0.54
BaAN ^a^	−55.501	−55.371	0.23	ANT ^a^	−69.617	−69.599	0.03	ANT ^a^	−64.852	−64.844	0.01
BANN ^a^	−60.793	−60.801	−0.01	BaAN ^a^	−91.938	−91.803	0.15	BANN ^b^	−97.418	−109.937	−12.85
BaP ^a^	−70.481	−70.748	−0.38	BaP ^a^	−97.864	−97.145	0.73	BaP ^a^	−103.536	−103.504	0.03
BbF ^a^	−44.365	−44.287	0.18	BbF ^a^	−103.742	−103.692	0.05	BbF ^a^	−100.331	−100.260	0.07
BeP ^a^	−67.026	−67.011	0.02	BghiP ^a^	−121.288	−121.378	−0.07	BeP ^a^	−74.155	−73.930	0.30
BghiP ^a^	−64.460	−64.604	−0.22	BkF ^a^	−112.373	−112.617	−0.22	BghiP ^a^	−98.467	−98.653	−0.19
BkF ^a^	−50.842	−50.926	−0.17	CHR ^a^	−83.232	−83.945	−0.86	BkF ^b^	−81.132	−60.361	25.60
CHR ^a^	−66.290	−65.886	0.61	CPPHN ^b^	−78.146	−66.951	14.33	CPPHN ^b^	−81.077	−45.480	43.91
CRN ^a^	−83.141	−83.186	−0.05	CRN ^b^	−113.449	−86.764	23.52	CRN ^a^	−86.995	−86.975	0.02
DahA ^b^	−66.480	−56.925	14.37	DahA ^a^	−105.255	−105.328	−0.07	DahA ^b^	−81.625	−62.226	23.77
FLR ^b^	−41.383	−46.911	−13.36	FLR ^a^	−67.423	−67.182	0.36	FLR ^b^	−83.469	−56.276	32.58
FRT ^a^	−49.042	−48.781	0.53	FRT ^b^	−104.230	−75.363	27.70	FRT ^a^	−76.780	−76.825	−0.06
IcdP ^a^	−65.071	−65.138	−0.10	NAP ^a^	−33.803	−33.696	0.32	NAP ^a^	−51.637	−51.298	0.66
NAP ^a^	−26.336	−26.624	−1.09	PHE ^a^	−54.440	−54.451	−0.02	PHE ^a^	−64.850	−65.097	−0.38
PHE ^a^	−34.993	−35.103	−0.31	PYR ^b^	−92.176	−75.580	18.00	PRL ^a^	−93.434	−93.337	0.10
PYR ^b^	−48.667	−53.742	−10.43					PYR ^a^	−89.548	−89.631	−0.09

Note: ^a^ training set; ^b^ test set.

## Data Availability

The data presented in this study are available contained within the article.

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
