# Peer review of "Potential Toxicity Risk Assessment and Priority Control Strategy for PAHs Metabolism and Transformation Behaviors in the Environment"

_ijerph, 2022, doi:10.3390/ijerph191710972_

Round 1
Reviewer 1 Report
Thank you for your extensive efforts of this study. I may understand the intention of authors, however unfortunately, I can understand almost nothing on how the authors proceed your study and thoughts to reach your intention.
Followings are my general comments for future consideration.
Abstract should be concise and tells only key methods and findings of the manuscript.
Authors reviewed literatures/models on variety of toxicities and transformations, but all should be considered in-depth knowledge of scientific way of thinking. One implication appears on a manuscript does not mean it is the general truth for another application. Authors should consider how existing findings can be integrated to your purpose and methodologies, carefully evaluating the applicability or limitations of each findings. I cannot find any of such real scientific discussion in the manuscript. Very sorry for you, but this is not the science.
Methods of the study should be much more clearly described. The text only lists the models, data, or information but I cannot understand what 3D-QSAR model was established through how integrating the listed sources and information.
Figure 4 and 5 discussed the “risk” of something, but what are the risk in the figures? This may be interesting, but the topic is worth huge discussions. At this appearance the figures 4 and 5 are completely nonsense from my understanding.
Reviewer 2 Report
This is an interesting article. However, my main concern is the length of the article, which is very long, and the style of the writing. Some sections seem to be mixed, making the manuscript difficult to follow. The authors should carefully consider each section and review the content, and eliminate information that it is already well known or add little value to their manuscript. For example, the first paragraph of the abstract can be shortened, or complete sections within the material and methods section should be displayed in the introduction instead. Please consider my comments below:
INTRODUCTION
Line 96. Please check the spelling of the organisms.
What is it new in this article? The aims are general, should be more specific. Authors should justify better why their study is relevant and what would add new to the current literature available on PAHs.
METHODS
Authors should check carefully this section and remove any information that it is not related to the methods. For example, lines 155-159. Consider also the information provided, for example, figure 2 – does it provide any information to the reader?
RESULTS & DISCUSSION
I think that table 3 is difficult to understand. Authors should provide a short description to facilitate the interpretation of this table. Explain why two different sets were also used. Same with Fig 3.
I consider that this section should be completely reviewed to eliminate any information that it is not related with the results or their discussion.
CONCLUSIONS
Conclusions should be also re-written to make them clearer and more specific to their study. Authors should consider rewrite line 591 to 606, as they do not describe any conclusion for their study.
Round 2
Reviewer 2 Report
Dear authors,
Thank you for addressing my comments. In relation to the spelling of "microorganisms" I was referring to their scientific names, but I see that you have removed them in your revised manuscript, so this is no longer needed. I think that the word organisms was spelling correctly in English before but this is something that you could review later. I would add a conclusive line in the abstract highlighting the major findings of your manuscript.
